# Effect of Chitosan Coating for Efficient Encapsulation and Improved Stability under Loading Preparation and Storage Conditions of *Bacillus* Lipopeptides

**DOI:** 10.3390/nano12234189

**Published:** 2022-11-25

**Authors:** Beom Ryong Kang, Joon Seong Park, Gwang Rok Ryu, Woo-Jin Jung, Jun-Seok Choi, Hye-Min Shin

**Affiliations:** 1Institute of Environmentally-Friendly Agriculture, Chonnam National University, Gwangju 61186, Republic of Korea; 2Gwangju Metropolitan City Agricultural Extension Center, Gwangju Metropolitan City 61945, Republic of Korea; 3Department of Agricultural Chemistry, Chonnam National University, Gwangju 61186, Republic of Korea

**Keywords:** chitosan, cyclic lipopeptide, liposome, nanoencapsulation

## Abstract

This study aims to evaluate the effect of chitosan coating on the formation and properties of *Bacillus* cyclic lipopeptide (CLP)-loaded liposomes. A nanoencapsulation strategy for a chitosan-coated liposomal system using lecithin phospholipids for the entrapment of antibiotic CLP prepared from *Bacillus subtilis* KB21 was developed. The produced chitosan-coated CLP liposome had mean size in the range of 118.47–121.67 nm. Transmission electron microscopy showed the spherical-shaped vesicles. Fourier transform infrared spectroscopy findings indicated the successful coating of the produced CLP-loaded liposomes by the used chitosan. Liposomes coated with 0.2% and 0.5% chitosan concentration decreased the surface tension by 7.3–12.1%, respectively, and increased the CLP content by 15.1–27.0%, respectively, compared to the uncoating liposomes. The coated concentration of chitosan influenced their CLP loading encapsulation efficiency and release kinetics. The physicochemical results of the dynamic light scattering, CLP capture efficiency and long-term storage capacity of nanocapsules increased with chitosan coating concentration. Furthermore, the chitosan-coated liposomes exhibited a significant enhancement in the stability of CLP loading liposomes. These results may suggest the potential application of chitosan-coated liposomes as a carrier of antibiotics in the development of the functional platform.

## 1. Introduction

Many biocontrol agents have been isolated and used to control plant pests, and products derived from *Bacillus* species make up the majority of commercially available biopesticides [1,2,3]. Their biocontrol effect has shown a direct antagonistic action against the pathogen inhibition through various metabolic materials. This pathogen proliferation inhibitory action is due to the production of bioactive substances and has been well reported as an important mechanism capable of inhibiting pathogens in the host [2,4]. Although many peptide antibiotics have been described in the past few decades, antimicrobial peptides derived from microorganisms are considered promising bioactive material candidates due to their broad antimicrobial spectrum, reduced toxicity, high biosurfactant activity, potential for environmentally friendly mass production, and reduced resistance to target organisms [5]. *Bacillus* strains synthesize a range of cyclic lipopeptides (CLP) with broad-spectrum antimicrobial properties. These metabolites are classified into three classes according to their structure: surfactin, iturin, and fengycin [2,6,7,8]. In addition, CLPs are composed of unique amino acids and the length and composition of fatty acid side chains provide structural diversity that influences physicochemical properties and biological activity. Furthermore, the composition and length of fatty acid side chains of CLP exhibit various physicochemical properties, which provide significant structural diversity influencing their biological activity [6]. Moreover, different lipopeptides compounds are usually co-produced for each CLP family [9]. Fengycin and iturin CLP inhibit the growth of pathogens with potent antimicrobial activity against a wide range of pathogens [7,10,11,12].

In recent years, various nanoencapsulation methods have been studied to effectively load proteins and natural products and protect them from the stimulated environments [13,14,15,16,17,18,19]. There is also a growing interest in using these methods to improve the delivery of bioactive substances to targeted sections. Among them, the encapsulation of bioactive substances is mainly made by methods such as spraying, freeze drying, extrusion, and emulsification [20]. In this study, we focused on obtaining chitosan-coated liposomes with better stability and active ingredient release properties in the inhibition of plant pathogens. Phospholipid liposome is a spherical lipid vesicle structure formed by one or more phospholipids surrounding an aqueous core. The liposomes are ideal carriers for the delivery of biologically active substances because of their low toxicity, self-assembly, and ability to bind hydrophilic and hydrophobic molecules. However, the practical application of liposomes has been limited by their poor physical and chemical stability during preparation and storage [21,22]. Another strategy to improve polymeric liposomes’ stability is to replace the phospholipid membrane with a membrane composed of amphiphilic polymers that can self-assemble; it also utilizes the interaction of lipid vesicles with oppositely charged polyelectrolytes to modify the liposome’s surface [16]. Coating lipid structures with polycations minimizes the release of the active ingredients from the unwanted sites and the cellular uptake of vesicles by cells due to the positive charge of the polycation coating. Here, it is worth mentioning that chitosan (CS) is the most widely studied as a stability-enhancing agent for coating liposomal surfaces due to its ability to increase the stability of vesicles, encapsulation efficiency, and surface adhesion properties [17,22,23]. In addition, amphiphilic CS derivatives have been studied as delivery systems for gene, protein, and natural products due to their solubility in organic and aqueous solvents and their ability to self-assemble under appropriate conditions [22,24,25].

The hypothesis of this study is that surface nanoengineering through biopolymer coating of polysaccharides improves the stability of liposomes. We speculated that the hydrophobic force between the CS and phospholipid would stabilize the liposomal membrane. Based on this hypothesis, we explored some unresolved problems. First, we focused on designing and evaluating CS-coated concentration as carrier for encapsulating and delivering the *Bacillus* CLP-loaded liposomes. It was our main interest to report on the successful encapsulation of the *Bacillus subtilis* KB21 (Bs KB21) lipopeptides (surfactin, iturin, and fengycin) in CS-coated liposomes. Hence, it may be synthesized with the physicochemical properties of the CS coating and the hydrophobic functional material properties of the liposome loaded with *Bacillus* CLP as a physiologically active material. Second, these developed carriers can effectively increase the stability of bioactive substances under stimulated environments. It was also important to characterize these self-assemblies, and evaluate the properties of CS coating concentration and CLP-loaded liposomes. The size distribution and morphological properties of the synthesized products were characterized by field emission transmission electron microscopy (FETEM), attenuated total reflection-Fourier transform infrared (ATRFTIR) spectroscopy, and zeta potential measurement. In addition, CLP entrapment efficiency was evaluated using liquid chromatography mass spectrometer (LC-MS/MS). The results of this study are expected to provide a new perspective for improving the biocontrol availability of antimicrobial active substances and expanding the potential application of antimicrobial agents in active ingredient delivery systems.

## 2. Materials and Methods

### 2.1. Materials

CS powder (91% degree of deacetylation, 16 cPs) derived from red snow crabs was purchased from Amicogen C&C, Co., Ltd. (Uljin-gun, Republic of Korea). Fat-free soybean lecithin with phosphatidylcholine (Lipoid S100, 94% soybean phosphatidylcholine, 3% lysophosphatidylcholine, 0.5% N-acyl-phosphatidylethanolamine, 0.1% phosphatidylethanolamine, 0.1% phosphatidylinositol, 2% water, and 0.2% ethanol) was purchased from Lipoid GmbH (Ludwigshafen, Germany) for the production of the liposomes. The absolute ethanol, high-pressure liquid chromatography grade methanol, acetic acid (99.9%), glycerol (99.9%), phosphate buffered saline (PBS), surfactin standard, iturin A standard, and fengycin standard were purchased from Sigma-Aldrich (Merck KGaA, Darmstadt, Germany).

### 2.2. Production of Antifungal Substance

Bs KB21 was cultured at 28 °C and 150 rpm using tryptic soy broth (TSB; Becton Dickinson GmbH, Heidelberg, Germany) medium. CLP purification was performed following the previously described procedure [26]. CLP produced by Bs KB21 was collected 4 days post-inoculation (dpi) from TSB medium. Bs KB21 was collected from the culture by centrifugation for 30 min at 8000× *g* at 4 °C, and the supernatants were filtered through a 0.22 µm pore size Millipore^®^ filter (Merck KGaA, Darmstadt, Germany) to remove the bacterial cells. The supernatant was further treated by acidification to pH 2.0 using a 6 M hydrochloric acid solution, and the acidified supernatant was left overnight at 4 °C for complete precipitation of the biosurfactant. The precipitate was centrifuged at 10,000× *g* for 20 min to pellet the crude biosurfactant. This pellet was then solvent-extracted with methanol. After removing the methanol, distilled water was added to dissolve the biosurfactant, and the CLP solution was stored at 4 °C.

To estimate the antifungal biosurfactant CLP production, the surface tension (mN/m) of the filtrates was measured in triplicate using the du Noüy Ring method with a surface tensiometer (K6; KRÜSS GmbH, Hamburg, Germany) at 25 °C [11]. A total of 10 mL of the filtrate was then placed on the tensiometer platform. The average value was used to express the surface activity of each sample. The negative control was sterile TSB medium without Bs KB21 cell-free supernatant at zero time. The surface tension of pure water was used to calibrate the instrument.

### 2.3. Preparation of the CS-Coated CLP-Loaded Liposomes

For the preparation of CS-coated CLP-loaded liposomes for nanovesicle encapsulation, 0.5% (*w*/*v*) lecithin was dissolved in ethanol at 40 °C using an ultrasonic bath at 20 kHz for 20 min. During this process, the temperature and magnetic stirring of the system were kept at 40 °C and 500 rpm, respectively. After the complete dispersion of lecithin in ethanol, 0.5% (*v*/*v*) of the microbial CLP solution was added to the phospholipid phase and stirred at 40 °C for 1 h until a homogeneous mixture was obtained. The resulting material was combined with the lecithin dispersion in ethanol. After this process, the final aqueous dispersions containing the formed CLP-loaded liposomes were cooled to 4 °C for the consolidation of the structures and their subsequent characterization. The CS-coated CLP-loaded liposome (CCL) was prepared by mixing the liposome suspension with an equal volume of CS solution. CS was dissolved in acetic acid (1% *v*/*v*) and stirred for 16 h at 25 °C to prepare a 1% (*w*/*v*) solution. The CLP-loaded liposome was then added to the 0, 0.2, and 0.5% (*v*/*v*) CS solutions while stirring at room temperature for 30 min (pH < 5.5) and then stored at 4 °C. The pH values of the solutions were adjusted with sodium hydroxide (1 M) and hydrochloric acid (1 M).

### 2.4. Characterization of the CCL

This study aims to develop functional crop protection products with good storage stability of their properties and active ingredients. Therefore, the structural properties of nanoencapsulated bioactive materials were confirmed through FETEM. In addition, the characteristics of dynamic light scattering (DLS), encapsulation efficiency, release properties and physical stability were determined.

#### 2.4.1. Dynamic Light Scattering, Fourier Transform Infrared Spectroscopy, and Surface Morphology of the CCL

The size distribution and zeta potential were determined via DLS using a Zetasizer Nano ZSP (Malvern Panalytical Ltd., Malvern, UK) with standard settings [27]. The stability of the liposomes was determined by monitoring changes in the size, zeta potential, and polydispersity index (PDI). FTIR spectroscopy analysis was done using a Vertex 70 v/Hyperion 2000 FTIR spectromicroscope (Bruker Corp., Munich, Germany). Samples were finely ground and placed in a diamond attenuated total reflection cell. The resolution was 4 cm^−1^, and 64 scans were signal-averaged in each interferogram over a wavenumber range of 4000–600 cm^−1^. The surface morphology of the nanoliposome encapsulation was evaluated by TEM (JEM-2100F, JEOL Ltd., Tokyo, Japan).

#### 2.4.2. Determination of Encapsulation Efficiency of the CCL

The encapsulation efficiency of the CCL was determined as described by Silva, et al. [18]. Encapsulation efficiency was calculated as the difference between the total amount of CLP used for encapsulation and the amount of residual CLP present in the supernatant after centrifugation of the encapsulated suspension using the following formula: Encapsulation efficiency = [(Total amount of CLP − residual amount of CLP)/Total amount of CLP] × 100.

The residual CLP of the iturin, surfactin, and fengycin CLP in the supernatant after centrifugation (8000× *g* for 10 min at 4 °C) was filtered using a 100 kDa AMICON^®^ Ultra-15 centrifugal filter (Merck KGaA, Darmstadt, Germany) to separate any unencapsulated CLP and remove free CS. The total amount of CLP used for encapsulation was determined using a LC-MS/MS (AB SCIEX Pty. Ltd., Framingham, MA, USA) with a capcell core C_18_ high pressure liquid chromatography column (2.1 × 150 mm, 2.7 μm; OSAKA SODA Co., Ltd., Osaka, Japan) at a flow rate of 0.3 mL/minute. To detect iturin, surfactin, and fengycin simultaneously, a binary solvent system was run in gradient mode. Methanol (A) and ultrapure water (B) containing 0.1% formic acid were used as mobile phases. The gradient was increased from initial 15% to 60% of solvent A within 1.5 min. After it was increased to 90% at 10 min, the mobile phase A was increased gently to 98% at 12.1 min, and then kept for 16 min, followed by a decrease to initial conditions of 15% A and held for 8.9 min to allow for equilibration. All compounds were ionized in electrospray ionization mode and analyzed in MRM (multiple reaction monitoring) mode (Appendix A). The compounds were identified and quantified by comparing their retention times and masses with those of commercial standards, as described by Kang, Park and Jung [7].

#### 2.4.3. In vitro CLP Release and Stability

To measure the release of CLP in vitro, CCL was suspended in PBS (pH 7.4) and incubated at 37 °C. The mixture was centrifuged at 10,000× *g* for 10 min at different time intervals (0, 0.5, 1, 1.5, 2, 3, 6, 9, 12, 24, 48, 60, and 72 h), and then the supernatant was removed and replaced with the same volume of PBS buffer [18]. At predetermined time intervals, samples were extracted and the CLP trapping efficiency of the dispersed liposomes were determined. The amount of released CLP iturin, surfactin, and fengycin CLP was determined by LC-MS/MS.

### 2.5. Statistical Analysis

All experiments were performed at least three times per treatment. Data are expressed as the mean ± standard errors. The data were analyzed using one-way analysis of variance, followed by Duncan’s multiple range tests (*p* < 0.05 was considered significant). All statistical analysis was performed using SPSS Version 23.0 (SPSS Inc., Chicago, IL, USA).

## 3. Results and Discussion

This study aimed to develop a novel nanoliposome platform based on self-assembling the polysaccharide chains (derived from CS) and *Bacillus* CLP with a hydrophilic group (amino acid or peptides; di- or polysaccharides; anions or cations) or a hydrophobic group (saturated or unsaturated fatty acid). To overcome the difficulties caused by the low solubility of CLP, crude CLP was extracted from *Bacillus* biosurfactants as the starting molecule. The CS derivatives were obtained by a simple chemical approach, and the resulting CCL was directly linked to the amino acid chain, significantly reducing the water solubility of the polysaccharides. Interestingly, this synthetic approach could be applied to negative, positive, or neutral charged polysaccharides.

The synthesized polymeric compounds were easily molded to obtain highly stable nanoliposome systems with antifungal effects that can be used in agricultural applications, particularly focusing on vesicle delivery. The procedure adopted in this study to form nanoliposomes is based on the ability of polysaccharide chains containing a hydrophobic moiety to self-assemble in nanostructures under specific experimental conditions.

### 3.1. Surface Tension and Mass Spectrometry Analysis of the CLP

The CLP levels of the prepared CCL products were analyzed using surface tension and LC-MS/MS. The surface tension was determined for the biosurfactants’ content after the complete hybridization process. The formation of the CLP nanostructures and their complete hybridized nanoliposome products used for the CLP synthesis was achieved using cultured Bs KB21 supernatants. As shown in Figure 1a, the surface tension was lower in the 0.5% (*v*/*v*) CCL with CS coating (26.2 mN/m) than in the culture filtrate products (33.0 mN/m). The surface tension decreased as the CS coating concentration increased at a constant lecithin level. In addition, CS-coated products (0.2 and 0.5% CCL) were significantly reduced compared to the cell-free supernatant (culture filtrate) and uncoated CS (0% CCL). This reduction showed the formation of CS-coated liposomes between the CLP and phospholipids, indicating complete hybridization to successfully form encapsulated nanoliposomes. Furthermore, this suggested that the CS coating resulted in changes in intramolecular and intermolecular interactions, i.e., hydrogen bonding and hydrophobic and electrostatic interactions, because of molecules’ conformational transition within the liposomes [27,28,29,30].

We previously reported that the CLP of Bs KB21 were identified as indistinguishable from the iturin A, surfactin, and fengycin CLP standards by mass spectrometry [11]. Therefore, the extracted CLP fractions from the CS-coated liposome products were analyzed by LC-MS/MS. As a result of the CLP quantification, the average CLP content increased with CS coating in the 0 to 0.5% concentration range, regardless of the phospholipid concentration in the liposomes (Figure 1b). For the 0.5% CCL, the CLP level was a total of 500.8 mg/kg, including purified surfactin, iturin, and fengycin CLP. In addition, the surfactin CLP was the highest among the CLP contents of all the CCL products. The data showed that the CCL contained 125.2–235.4 mg/kg surfactin, 49.4–141.9 mg/kg iturin, and 34.2–123.6 mg/kg fengycin CLP. Among the three CLP biosurfactant families, surfactin has the most important surface activity. According to Deleu, et al. [31], surfactin (100 mg/L) decreased the surface tension between dodecane-in-water to 2.45 mN/m versus 16.3 and 13.5 mN/m for iturin A and fengycin, respectively. It has been reported that the most efficient biosurfactants can reduce the surface tension of water from 72 to less than 30 mN/m and the surface tension of water/oil systems from 43 to less than 1 mN/m [32]. Moreover, our previous study’s results showed that the surface tension of the Bs KB21 supernatant reduced from 74.1 to 29.1 mN/m within 24 h of incubation and remained constant after that up to 216 h of fermentation [11].

### 3.2. Morphology, Diameter, Polydispersity Index, and Zeta Potential

The TEM image of CCL prepared by lecithin phospholipid is presented in Figure 2; they appear as nanometric-sized and spherical-shaped liposomes. The core-shell structure of the CCL displayed a darker core due to negative staining of the encapsulated CLP along with the CS, which could be attributed to lecithin entrapment. TEM imaging revealed a core shell nanosystem with complete and uniform coating on the nanoencapsulation surface (Figure 2).

The DLS results, which concerned the nanometric population, are summarized in Table 1. The average diameter of the CCL in the 0.5% CCL was 121.67 ± 2.23 nm (Figure 3). The PDI was 0.25 ± 0.02 and the zeta potential value was −10.47 ± 0.44 mV (Table 1). The mean diameter value of the 0.5% CCL was increased compared to the uncoated CS (0% CCL). In particular, the increasing of CS coating concentration significantly increased the Z-average values of the liposomes without aggregation between vesicles, demonstrating that the added CS covered the vesicles [15,33]. It is also known that the tendency to aggregate during liposomes formation increases with higher lipid concentrations [34]. For the CCL products, the PDI values were lower than those of the culture filtrates or CS-uncoated liposomes. The PDI values of the CS-coated liposome formulations were less than 0.29, whereas the uncoated samples were more than 0.38 (Table 1). These high PDI values indicate a broad size distribution and may be due to the inclusion of large unbound particles or aggregates (Figure 3) [35]. Therefore, the most notable features of the CS coating in our results are its stability despite the absolute value decrease in the zeta potential and a low PDI value indicating a monomodal particle size distribution.

The zeta potential is an important indicator to measure the surface charge and is used to predict and control the electrostatic stability of colloidal suspensions. Our study showed that the zeta potential increased to a constant level with the addition of the CS coating layer (Table 1). When the CCL was present, the zeta potentials increased, suggesting that some cationic CLP could interact with the surface. This change in surface charge was most likely due to the ionic attraction between the positive charged CS amino groups and the negative charged CLP-loaded liposome surfaces and was indicative of the successful coating of CS onto the CLP-loaded liposome surface [36,37]. In addition, the particles dispersed in an aqueous medium tends to acquire a surface electrical charge, which is generally thought to be due to changes in the diffuse layer with the CLP dispersed in the continuous phase. Therefore, it could be determined that the CLP-loaded was located both inside and on the liposome’s surface.

Moreover, this negative charge density came from the CLP-loaded complexed liposomes and is thought to be balanced by the charge-shielding effect of negative chains [38]. Therefore, the various mixing conditions used during liposome preparation can result in distinct vesicle assembly patterns that affect their final characteristics [39]. In this study, we used parameters that could influence the formation of vesicles such as Lipoid S100 lecithin, by the concentration, ratio, and stirring speed of the ethanol. The optimized vesicle concentrate contained 10% (*w*/*w*) of lecithin dissolved in 30% (*w*/*w*) of ethanol. When the vesicle concentrate was mixed with water, the resulting mixture contained the concentrate disrupted into small-dispersed units. Following the gradient of concentration, the ethanol diffused from the large surface of the small units to the surrounding water. The remaining lecithin precipitated almost immediately in the non-solvent environment as small aggregates. This process is supported by lecithin types with a high amount of phosphatidylcholine [40]. Therefore, Lipoid lecithin was used for liposomes with high aggregation potential. The low zeta potential was associated with the high content of phosphatidylcholine to form a liposome system with low electrostatic repulsion [41], which resulted in liposomes with a larger diameter.

### 3.3. Encapsulation Efficiency and In Vitro Release Kinetics of Bs KB21 CLP

As previously mentioned, the CLP encapsulation efficiency was calculated as a percentage of each CLP type in the total CLP content. Increased encapsulation efficiency was observed with increasing CLP and CS coating concentration. The encapsulation efficiency of 0.5% CCL was 81.4%, the highest for all the formulations tested (Table 1). This could be because the unsaturated fatty acyl chains of lecithin phospholipid provide high membrane flexibility, allowing more CLP to be entrapped into the liposomes [40]. These results showed that CLP compounds were efficiently loaded into the liposomes with an improved encapsulation efficiency. Moreover, the fengycin CLP of the 0.5% CCL showed higher encapsulation efficiency (87.8%) than those of the surfactin (78.2%) and the iturin CLP (81.7%) (Figure 4a). Fengycin and iturin are surface-active agents with both lipophilic and hydrophilic moieties that present broad antifungal activity. Fengycin CLP showed higher encapsulation efficiency than the other CLP as the CS coating concentration increased. This result could be related to the charge of the encapsulated compounds; negative charged proteins or CLP can be attracted to positive charged CS allowing high encapsulation efficiency [19]. Our results were consistent with other compounds encapsulated in liposomal CS/CLP prepared by ionotropic hybridization. It has been previously reported that bacillomycin D, surfactin CLP, and daptomycin showed significant encapsulation efficiencies ranging from 80.8% to 97.9% [18,19]. Hydrophobic cyclosporine A was reported to exhibit encapsulation efficiency of 73–85% when loaded into CS [42]. In addition, the increase in CLP encapsulation efficiency for CS was shown to be closely related to their concentration [43,44]. The 2:1 CLP:CS ratio showed the best encapsulation efficiency. However, increasing the CLP concentration ratio to three times, a three-fold decrease in encapsulation efficiency was observed. Therefore, this study could establish the hypothesis that surface nanoengineering through biopolymer coating of polysaccharides could improve the stability of liposomes.

To explore the release profile of the CLP biosurfactants from the formulations under physiological conditions with different CS coating concentration, in vitro CLP release experiments were conducted. As shown in Figure 4b, the release profile of total CLP from uncoated CS liposomes (0% CCL) revealed a continuous release of CLP after 3 h of incubation, reaching 47.5% release. After 24 h of incubation, only 84.8% of the total CLP had been released. In comparison, 0.5% CCL was released in a low and continuous manner reaching 22.4% after 3 h and 71.7% after 24 h. In addition, 0.5% CS-coated liposomes showed weaker release than 0.2% CCL, 11.1% after 3 h and 6.9% after 24 h. When the CLP-loaded liposomes were coated with CS, the CS concentration had a significant effect on CLP release. The half level for total CLP release were 6, 9, and 12 h, respectively, when the CS coating ratio was increased from 0 to 0.5%. Thus, the concentration of 0.5% CS had a minor effect on the total CLP release properties of the liposomes. At 52 h, >90% of the total CLP in 0.5% CCL had been released in a sustained manner. Surfactin CLP containing two negative charged amino acid residues (Glu and Asp) should be attracted to CS compounds resulting in high binding efficiency. However, the release of surfactin CLP from the 0.5% CCL was greater than for the iturin and fengycin CLP. This observation is attributable to the presence of vesicular structures in the liposomal formulation that reduced the release rate. A slow release of total CLP was observed at 0.5% CCL, which is attributed to the CS coating. These results could be explained by previous results that proteins with negatively charged surfaces are loaded more efficiently in CS nanoparticle and are released more slowly [19,45].

### 3.4. Fourier Transform Infrared Analysis

FT-IR spectrometry was used to study the composition and structural properties of the components and the nanocomposites (Figure 5). The results confirmed that the structural features of CS were represented by a broad band at 3350 cm^−1^, which was assigned to the –N-H stretching vibration mode superimposed on the –O-H stretching vibration mode. The peak at 2870 cm^−1^ was attributed to the –C-H stretching vibration band. A characteristic peak at 1650 cm^−1^ was associated with the vibration of the –C=O bonds of the amide group, while the –NH bending vibration was shown at 1585 cm^−1^. The peaks observed at 1150 and 1030 cm^−1^ are related to the vibrations of the –C-O groups.

The filtrates of Bs KB21 showed a broader absorption peak at 3315 cm^−1^ in the presence of –OH or –NH groups. Peaks at 2926, 2854, 1454, and 1402 cm^−1^ confirm the –C-H stretching (–CH_3_ and –CH_2_) of the aliphatic chain of the lipid. A similar stretching for –C-H of lipid has been previously reported [46]. The presence of N-H bending of secondary amides (1544 cm^−1^) and the carbonyl group (C=O) of amide (1664 cm^−1^) confirmed the peptide parts. Peaks at 1242 and 1076 cm^−1^ are probably due to C-O-C vibrations in esters. The FTIR spectrum of the biosurfactant showed similarity to the CLP, such as surfactin, iturin, and fengycin produced by Bacilli, as reported in other studies [47]. The antifungal substances produced by Bs KB21 were thus identified as CLP compounds. In addition, the appreciable differences between spectra of CCL and CS/CLP are observable. As shown in Figure 5, the intensity of the C=O vibration band in the CS amide group (~1650 cm^−1^) was increased after CLP loading, while bands between 840 and 1150 cm^−1^ faded. New or shifting peaks at 1442, 1493, 1549, 1641, 2869, and 2970 cm^−1^ indicated interactions between the CS/CLP and phospholipid in the liposome; this showed that the CS/CLP was efficiently incorporated into the lipid liposomes.

## 4. Conclusions

Comparative analysis of the CLP-loaded liposomes derived from Bs KB21 cultures in the presence of a CS coating layer revealed the process of encapsulation by compositional and structural properties changes of the components. The CS coating improves the stability of CCL despite the lower zeta potential in the monomodal of particle size distribution. CLP expression by efficient integration of CS/phospholipids is expected to contribute significantly to the antifungal activity. Furthermore, the morphological changes observed in this study strongly supported the relevant role of the CS coating as a key factor in the antifungal bioactivity of the CLP-loaded liposomes. These studies complemented the fact that the CS constituent substrates are important factors determining the quantitative and qualitative levels of the bioactive metabolites during liposome synthesis. Therefore, applying functionalized, safe CS-coated liposomes to encapsulate small traces of compatible microbial antibiotics can result in potent antimicrobial agents and provide a promising solution to mitigate the problem of drug resistance toward target organisms, and avoid the spread of infections by the pathogens.

## Figures and Tables

**Figure 1 nanomaterials-12-04189-f001:**
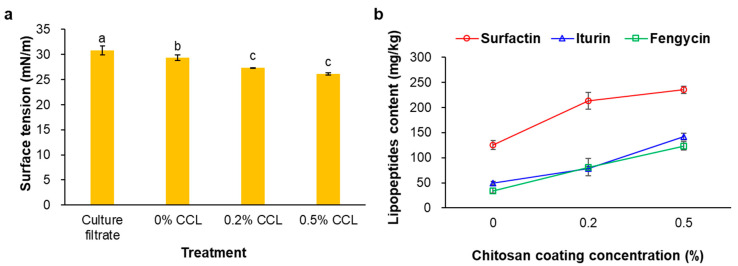
Surface tension (**a**) and CLP level (**b**) of the prepared CCL. The surface tension of water control was 73.2 ± 0.1 mN/m. Data represent means ± standard errors of three replicates. The values followed by different letters in a column were significantly different at *p* < 0.05 by Duncan’s multiple range test.

**Figure 2 nanomaterials-12-04189-f002:**
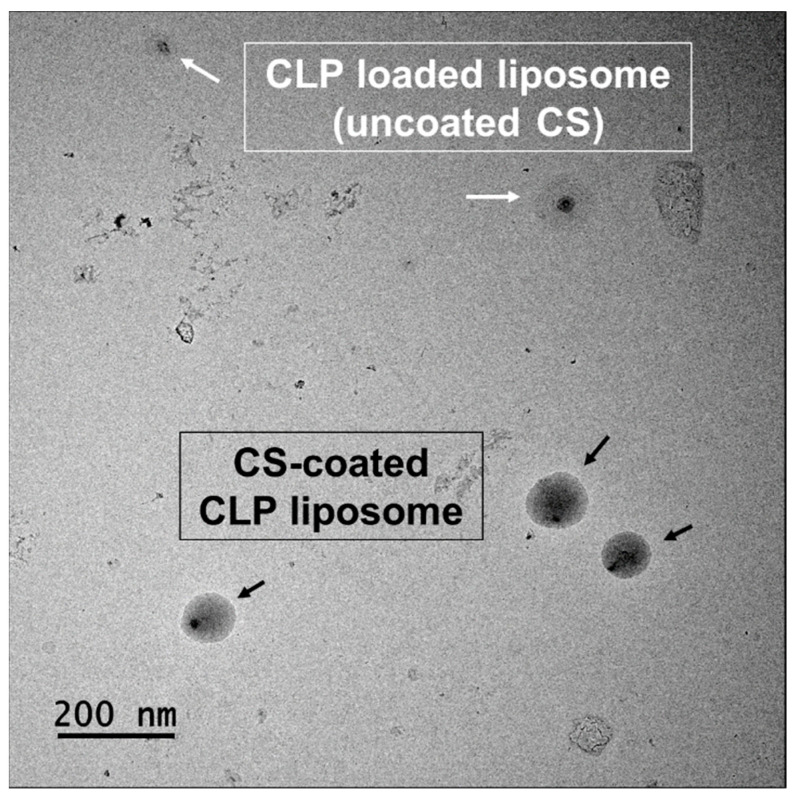
TEM image of CLP-loaded liposomes and CS-coated CLP liposomes (magnification = 25,000×).

**Figure 3 nanomaterials-12-04189-f003:**
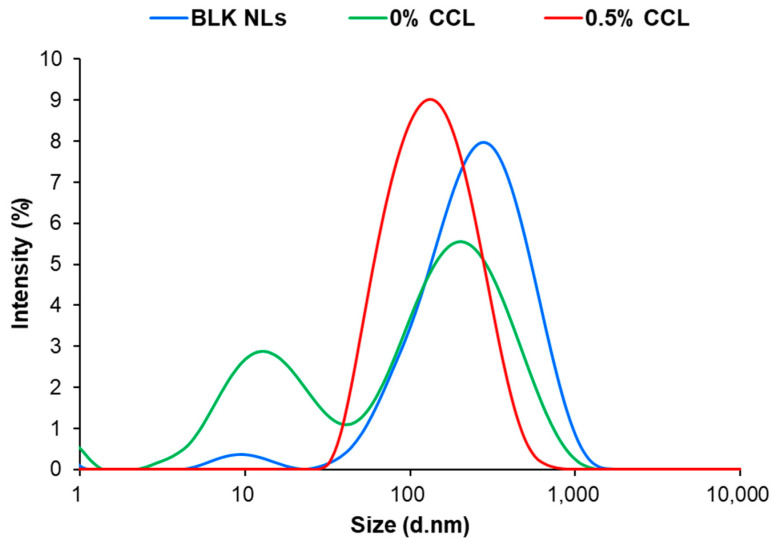
Size distributions (average diameter) of CS-coated CLP-loaded liposome (CCL). BLK NLs, blank liposome.

**Figure 4 nanomaterials-12-04189-f004:**
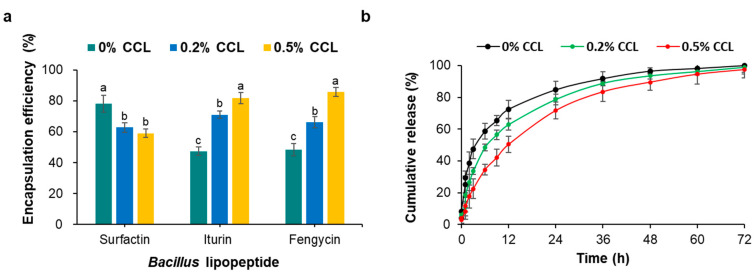
Encapsulation efficiency of CLP-loaded liposomes from CCL products (**a**) and in vitro the CLP release profiles (**b**) at various CS-coated concentration from CCL products. Data represent means ± standard errors of three replicates. The values followed by different letters in a column were significantly different at *p* < 0.05 by Duncan’s multiple range test.

**Figure 5 nanomaterials-12-04189-f005:**
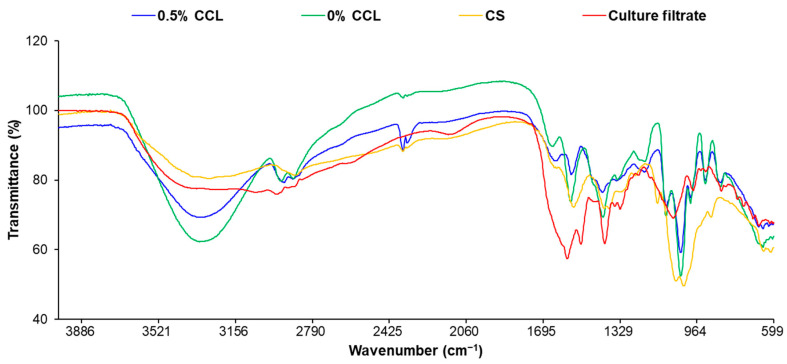
FT-IR spectra of 0% and 0.5% CCL, CS, and culture filtrate of Bs KB21.

**Table 1 nanomaterials-12-04189-t001:** Average diameter, polydispersity index, zeta potential, and encapsulation efficiency of chitosan (CS)—coated cyclic lipopeptide (CLP)—loaded liposome (CCL) on 0.5% lecithin phospholipid.

Treatment	Z-Average (nm)	Polydispersity Index	Zeta Potential (mV)	Encapsulation Efficiency (%)
Blank liposomes	72.83 ± 4.72 ^d^	0.22 ± 0.04 ^e^	−44.10 ± 2.50 ^c^	-
0% CCL (uncoated CS)	83.45 ± 3.21 ^c^	0.38 ± 0.02 ^d^	−27.30 ± 1.48 ^b^	54.0 ^b^
0.2% CCL	118.47 ± 6.38 ^b^	0.29 ± 0.03 ^c^	−11.77 ± 0.71 ^a^	63.4 ^b^
0.5% CCL	121.67 ± 2.23 ^b^	0.25 ± 0.02 ^b^	−10.47 ± 0.44 ^a^	82.5 ^a^
Culture filtrate	196.00 ± 7.60 ^a^	0.44 ± 0.01 ^a^	−48.04 ± 0.25 ^d^	-

The values followed by different letters in a column were significantly different at *p* < 0.05 by Duncan’s multiple range test. Data represent mean ± standard error of three replicates.

## Data Availability

Not applicable.

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
