# Peer review of "Effect of Chitosan Coating for Efficient Encapsulation and Improved Stability under Loading Preparation and Storage Conditions of Bacillus Lipopeptides"

_nanomaterials, 2022, doi:10.3390/nano12234189_

Round 1

Reviewer 1 Report

The manuscript entitled "Effect of chitosan coating for efficient encapsulation and improved stability under loading preparation and storage conditions of Bacillus lipopetides" by Beom Ryong Kang et al embodies a good amount of experimental data concerning the chitosan (CS) encapsulation (coating) of liposomes containing cyclic lipopeptides (CLL). The aim of this endeavor is to obtain chitosan coated liposomes containing lipopeptides (CCL) with better stability and active ingredient release properties in the inhibition of plant pathogens.

There are several zones of the manuscript that need rephrasing due to confusing and/or elliptic formulations. The Conclusions may also get some improvement. All specific observations and suggestions are to be found in the attached reviewed form of the manuscript.

Author Response

Thank you for your kind and detailed review of the results. Recommended corrections have been made in the text. The revised thesis is attached.

Reviewer 2 Report

This manuscript atempts to describe the benefits that can possibly be obtained upon chitosan coating for the encapsulation of Bacillus cyclic lipopeptide, in order to decrease the crucial problem of drug resistance. 

The paper is very well written and the observed data are clearly described by means of  simple  coloured graphics. TEM and ZP techniques were employed to characterize the size and external charges of liposomes. ATR-FTIR has also provides interesting information on the composition of coated liposomes. 

I can only suggest some minor changes to this manuscript: (i) Please do not use abbreviations (such as FETEM and FT-IR in Abstract. (ii) Conclusion (Section 4) is sufficiently clear, and does not really need to refer again to Tables,Figures and already cited reference [3] . All these informations can be found in the preceding Section devoted to the description of Results. (iii) In order to better understand the usefulness of FT-IR spectra, it would have been better to transfer the IR peak positions from the Section 3.5 to an extra Table, along with their tentative assignments, gaining place in the text to describe the apparition/suppression of certain bands and the reasons behind these changes.    

Author Response

  1. Please do not use abbreviations (such as FETEM and FT-IR in Abstract.

Response. Recommended correction was made in the text; Transmission electron microscopy showed the spherical-shaped vesicles. Fourier transform in-frared spectroscopy findings indicated the successful coating of the produced CLP-loaded lipo-somes by the used chitosan.

  1. Conclusion (Section 4) is sufficiently clear, and does not really need to refer again to Tables,Figures and already cited reference [3] . All these informations can be found in the preceding Section devoted to the description of Results.

Response. Recommended correction was made in the text;

Comparative analysis of the CLP-loaded liposomes derived from Bs KB21 cultures in the presence of a CS coating layer revealed the process of encapsulation by compo-sitional and structural properties changes of the components. The CS coating improves the stability of CCL despite the lower zeta potential in the monomodal of particle size distribution. CLP expression by efficient integration of CS/phospholipids is expected to contribute significantly to the antifungal activity. Furthermore, the morphological changes observed in this study strongly supported the relevant role of the CS coating as a key factor in the antifungal bioactivity of the CLP-loaded liposomes. These studies complemented the fact that the CS constituent substrates are important factors deter-mining the quantitative and qualitative levels of the bioactive metabolites during lipo-some synthesis. Therefore, applying functionalized, safe CS-coated liposomes to en-capsulate small traces of compatible microbial antibiotics can result in potent antimi-crobial agents and provides a promising solution to mitigate the problem of drug re-sistance toward target organisms, and avoid the spread of infections by the pathogens.

  1. In order to better understand the usefulness of FT-IR spectra, it would have been better to transfer the IR peak positions from the Section 3.5 to an extra Table, along with their tentative assignments, gaining place in the text to describe the apparition/suppression of certain bands and the reasons behind these changes.

Response. Reviewer comments are quite reasonable. We also tried it with a table. However, since the difference in analysis according to the wavelength of the sample was varied, all wavelengths were displayed.

Reviewer 3 Report

See comments on pdf file.

Author Response

Thank you for your kind and detailed review of the results. The content of the review has been corrected and attached.
